# Later Growth Cessation and Increased Freezing Tolerance Potentially Result in Later Dormancy in Evergreen Iris Compared with Deciduous Iris

**DOI:** 10.3390/ijms231911123

**Published:** 2022-09-22

**Authors:** Tong Xu, Jiao Zhang, Lingmei Shao, Xiaobin Wang, Runlong Zhang, Chenxi Ji, Yiping Xia, Liangsheng Zhang, Jiaping Zhang, Danqing Li

**Affiliations:** 1Genomics and Genetic Engineering Laboratory of Ornamental Plants, Department of Horticulture, College of Agriculture and Biotechnology, Zhejiang University, Hangzhou 310058, China; 2Department of Environmental Science and Landscape Architecture, Graduate School of Horticulture, Chiba University, Chiba 271-0092, Japan

**Keywords:** artificial dormancy induction, iris, phytohormone, carbohydrate, stress response, gene expression

## Abstract

Winter dormancy is a protective survival strategy for plants to resist harsh natural environments. In the context of global warming, the progression of dormancy has been significantly affected in perennials, which requires further research. Here, a systematic study was performed to compare the induction of dormancy in two closely related iris species with an ecodormancy-only process, the evergreen *Iris japonica* Thunb. and the deciduous *Iris tectorum* Maxim. under artificial conditions. Firstly, morphological and physiological observations were evaluated to ensure the developmental status of the two iris species. Furthermore, the expression patterns of the genes involved in key pathways related to plant winter dormancy were determined, and correlation analyses with dormancy marker genes were conducted. We found that deciduous iris entered dormancy earlier than evergreen iris under artificial dormancy induction conditions. Phytohormones and carbohydrates play roles in coordinating growth and stress responses during dormancy induction in both iris species. Moreover, dormancy-related *MADS-box* genes and *SnRKs* (*Snf1-related protein kinase*) might represent a bridge between carbohydrate and phytohormone interaction during iris dormancy. These findings provide a hypothetical model explaining the later dormancy in evergreen iris compared with deciduous iris under artificial dormancy induction conditions and reveal some candidate genes. The findings of this study could provide new insights into the research of dormancy in perennial plants with an ecodormancy-only process and contribute to effectively managing iris production, postharvest storage, and shipping.

## 1. Introduction

Winter dormancy is a protective survival strategy for plants to resist harsh natural environments [1] and is generally divided into two types: endodormancy, and ecodormancy [2]. Specifically, ecodormancy is a state induced by the limitation of external factors, such as temperature and light. At this stage, the buds can reacquire the ability to grow immediately under favourable environmental conditions, whereas this phenomenon is not observed in endodormant plants. A certain period of chilling accumulation (chilling requirement) leads to endodormancy release. Once endodormancy is broken, buds enter ecodormancy and relatively warm temperatures induce bud burst [3,4]. In the context of global warming, the progression of dormancy in perennials has been significantly affected. Specifically, the germination, growth, flowering and fruiting are directly affected in the following year, which results in reduced crop yield and loss of economic benefits [5,6]. On the other hand, dormancy can prevent premature germination of plants, ensure product consistency, prolong storage and ensure safe transportation [7,8,9]. Therefore, elucidating the mechanism regulating winter dormancy is of great significance for the commercial production of crops.

Winter dormancy in subtropical plants differs quite drastically from that in boreal/temperate regions because winter is shorter and warmer in subtropical regions compared with boreal and temperate regions [10]. Current studies on the mechanisms underlying winter dormancy mainly focus on perennial plants with endodormancy and ecodormancy [11,12,13], which are native to temperate and cold zones. In contrast, much less is known about the dormancy characteristics of perennials in the transition zone between temperate and subtropical climates. In this zone, many evergreen monocots, such as *Hemerocallis fulva* L., *Hippeastrum* spp., and *Iris* spp., only exhibit ecodormancy, showing dormancy when under stress but continued growth once conditions are favorable [7,14,15,16]. The underlying physiological and molecular underpinnings of winter dormancy in these plants have not been thoroughly investigated to date. Hence, it is important to uncover the regulatory mechanisms of winter dormancy in perennials exclusively exhibiting ecodormancy. Such studies contribute to fully elucidating the dormancy mechanism of perennials, expanding the planting range of crops and ensuring the safety of crop production, and aiding researchers in solving the problems caused by global warming [17,18].

Irises, one of the most popular herbaceous perennials, play an important role in the ornamental plants industry, perfume and cosmetics industry, landscaping and medicines. [19,20,21,22]. In previous studies, we found that irises did not require low temperatures to break dormancy during the overwintering process and only exhibit ecodormancy in the transition zone between temperate and subtropical climates [15]. *Iris japonica* Thunb. (IJ) and *Iris tectorum* Maxim. (IT) are closely related species that have contrasting winter dormancy durations and manifested as distinct evergreen or deciduous phenotypes in winter [23]. Therefore, these two irises are excellent materials for the study of winter dormancy and underlying mechanisms in perennials exclusively exhibiting ecodormancy. Under artificial conditions, environmental parameters can be regulated to create specific conditions for the growth and development of plants [24], avoiding interference from other factors in the natural environment. Additionally, experiments can be performed under artificial conditions without time and space restrictions, given that the climatic environment where the plant is located is reproducible [25]. Under natural conditions, deciduous irises in Hangzhou (China) show significant dormancy in November and December [15], and our previous experiment, which simulates air temperatures and photoperiod in November of Hangzhou, induced obvious morphological differences in two irises [26]. Thus, we artificially simulated the temperature and light conditions of November in Hangzhou to induce dormancy in IJ and IT so that the experiment is repeatable and not limited by natural conditions such as seasons.

Ecodormancy is a response mechanism that enables perennial plants to resist multiple abiotic stresses in winter, with no visible signs of growth [2,12]. Phytohormones are thought to be the main regulators during the transition progression between plant growth and dormancy [1], of which abscisic acid (ABA) is especially involved in abiotic stresses and dormancy [27]. In autumn, increased ABA levels and decreased gibberellin (GA) levels result in growth cessation, apical buds set, and induction of bud dormancy [28,29]. In addition, many studies have shown that carbohydrates are closely related to dormancy [5,30]. Starch is an important carbohydrate reserve for plants during dormancy [31]. Soluble sugars are involved in maintaining the normal function of cell membranes, enhancing cold tolerance, and acting as signal molecules regulating plant growth [30]. Over the past decade, our knowledge of the winter dormancy process at the molecular level has increased considerably based on advances in molecular biology and bioinformatics. Some *MADS-box* genes, such as the *Dormancy-associated MADS-box* genes (*DAMs*), *fruitfull* (*FUL*), and *flowering locus c* (*FLC*), have been identified as candidate genes controlling winter dormancy in perennial species [12,13,32,33]. In poplar (*Populus tremula* L.), ABA promotes the expression of *short vegetative phase-like* (*SVL*), which subsequently positively regulates ABA synthesis [34]. Moreover, GA biosynthesis and degradation pathways were demonstrated to be controlled by SVL encoding genes. These genes inhibit the expression of *GA 20-oxidase 1* (*GA20ox1*) and *GA20ox2* and promote the expression of *GA2-oxidase 8* (*GA2ox8*), preventing the accumulation of GA and thus inhibiting the release of dormancy [35,36,37]. In addition, grape (*Vitis vinifera* L.) *alpha-amylase* genes (*VvAMYs*) and leafy spurge (*Euphorbia esula* L.) *beta-amylase* genes (*EeBAM1*) were induced in dormant buds [38,39]. Overexpression of *sucrose phosphate synthase* (*SPS*) in poplars accelerated bud break due to increased accumulation of sucrose and starch compared with wild-type poplars. These results suggest that enhanced sugar and/or starch reserves may be an important factor regulating the release of dormancy [40]. However, related physiological and molecular changes in plants exclusively exhibiting ecodormancy remain largely unknown.

In this study, a systematic study was performed to compare the induction of ecodormancy in the closely related iris species, the evergreen IJ and the deciduous IT under artificial conditions. Firstly, morphological and physiological observations were evaluated to ensure the developmental status of the two iris species. Furthermore, the expression patterns of the genes involved in key pathways related to plant winter dormancy were determined, and then their correlation analyses with dormancy marker genes were conducted. Finally, we proposed a mechanistic model for investigating the key factors contributing to later dormancy in the evergreen iris relative to the deciduous iris. Our findings may lay a foundation for the further study of molecular mechanisms in perennials exclusively exhibiting ecodormancy in the transition zone between temperate and subtropical climates and provide a reference for plant postharvest storage, shipping and breeding.

## 2. Results

### 2.1. Dormancy Status Determination

To explore the regulatory mechanisms of winter dormancy in irises, we exposed the evergreen and deciduous iris species to artificial dormancy-inducing conditions (Table 1). Growth cessation is the first indication of dormancy induction [41]. The shoot height decreased gradually and then stabilized over time in IT, whereas that of IJ increased slowly (Figure 1A,B). Notably, obvious differences in the growth status of the two iris species were noted since Tre.4. The shoot height of IT tended to be stable, indicating a state of growth cessation, whereas IJ was still in a slow-growth trend at a growing state. In addition, the number of green leaves of IT decreased under the dormancy induction conditions, whereas no significant change was found in the number of green leaves of IJ (Figure 1C).

To judge whether plants enter the dormant state at the molecular level, *CYCD1;1*, *CDKB2;2*, *SVP2*, *FLC1*, *FLC2*, and *FUL2* (Gene full names are presented in Appendix A) were chosen as markers. The results showed that in both species, the expression of *CYCD1;1*, *FLC1*, and *FLC2* decreased during the dormancy-inducing conditions, whereas *SVP2* and *FUL2* expression increased (Figure 1D). Interestingly, transcript levels for *CYCD1;1*, *CDKB2;2*, *FLC1*, and *FLC2* increased from Tre.4 to Tre.5 in IJ, suggesting that IJ still had the tendency to grow at this stage. According to morphological observations and the expression of marker genes, Tre.1 to Tre.3 were classified as the dormancy induction stage, whereas Tre.4 to Tre.6 were considered as the dormancy maintenance stage for IT. In contrast, IJ was at the dormancy induction status during the entire experiment. These results indicated that IT entered a dormant state earlier than IJ under our artificial dormancy induction conditions.

### 2.2. Changes in ABA and GA_3_ Levels and Related Gene Expression

To further dissect the differences between evergreen and deciduous irises during dormancy, sampling stages when morphological differences occurred (Tre.1, Tre.4, Tre.5 and Tre.6), were selected for further experiments. We measured ABA and GA_3_ levels in apical shoot and leaf tissues of both iris species during these stages. In the apical shoot tissues, the ABA level was high during dormancy induction, and increased during dormancy maintenance stage in IT (Figure 2A). Similarly, the ABA level of IJ presented an increasing trend during dormancy induction. In the leaf tissues, ABA in IT was at a low level during dormancy induction and was maintained at relatively high levels during the dormancy maintenance stage (Figure 2B). However, the ABA level fluctuated during dormancy induction in IJ.

GA_3_ had high levels in IT and exhibited a downward trend in IJ in the apical shoot tissues during dormancy induction (Figure 2C). In the apical shoot tissues of IT, a gradual increase was observed during the dormancy maintenance stage. GA_3_ level in IT was low and presented an overall downward trend in IJ during dormancy induction in the leaf tissues (Figure 2D). During the dormancy maintenance stage, the GA_3_ level of IT first increased and then decreased in the leaf tissues.

To better understand the mechanisms linked to the phytohormone signalling pathway during dormancy, we determined the expression of genes involved in ABA and GA_3_ biosynthesis, catabolism, and signal transduction pathways in the apical shoot tissues. The ABA synthesis-associated genes *NCED1*, *ABA2*, and *AAO3* and *CYP707A* genes involved in ABA catabolism had low expression levels during dormancy induction in IT. During dormancy maintenance stage, these ABA synthesis-associated genes was highly expressed in early stage and then downregulated in IT. Meanwhile, the expression of *CYP707A* was also downregulated at this stage in IT. The expression of ABA synthesis-associated genes was upregulated during dormancy induction, while *CYP707A* decreased until Tre.5 during dormancy induction in IJ. Additionally, ABA synthesis-associated genes in IT were upregulated earlier than those in IJ, which was consistent with IT entering a dormant state earlier than IJ. In the ABA signal transduction pathway, *PYL8*, *SNRK*, *BZIP12* and *ABF1* were upregulated, whereas *PP2C5* was downregulated during dormancy maintenance stage in IT and dormancy induction in IJ (Figure 2E).

*GA20ox2* is a key GA biosynthesis gene. However, *GA2ox2* is a key GA catabolism gene. Their expression patterns were consistent with changes in GA_3_ levels in IT (Figure 2E). In contrast, *GA20ox2* was upregulated, whereas *GA2ox2* expression first increased and then decreased during the dormancy induction stage in IJ. The GA signal transduction-related gene *SPY* presented a contradictory trend as the GA_3_ levels changed. In addition, *RGA* and *GAI* encoding the DELLA protein, which connects the ABA and GA signal transduction pathways [41], were downregulated during dormancy maintenance in IT but upregulated during dormancy induction in IJ, and similar PIF expression trends were noted in both species.

### 2.3. Changes in Carbohydrate Content and Related Gene Expression

In the apical shoot tissues, the contents of starch decreased at the transition from dormancy induction to the dormancy stage and increased during dormancy maintenance stage in IT, whereas no significant change was noted in IJ during dormancy induction (Figure 3A). Soluble sugar (sucrose, glucose and fructose) contents were upregulated from dormancy induction to the dormancy stage, whereas were downregulated during dormancy maintenance stage in IT (Figure 3B–D). During dormancy induction, soluble sugar contents displayed upward trends in IJ (Figure 3B–D). Notably, an inverse relationship developed between starch and soluble sugar contents in IT during the shift from dormancy induction to the dormancy stage and continued through the dormancy maintenance stage. This phenomenon was not observed in IJ (Figure 3A–D).

In the leaf tissues, unlike the observations in the apical shoot tissues, starch content was at a low level in IT and increased in IJ during dormancy induction (Figure 3E). During dormancy maintenance stage, starch content was maintained at relatively high levels in IT (Figure 3E). Soluble sugar contents presented similar trends in apical shoot tissues, which were elevated during dormancy induction in IJ (Figure 3F–H). However, soluble sugar contents increased during the dormancy maintenance stage in IT, which presented opposite trends in the apical shoot tissues (Figure 3F–H). Interestingly, the increasing trends of sucrose in IT occurred earlier than those in IJ in both apical shoot and leaf tissues (Figure 3B,F). These findings are consistent with IT entering a dormant state much earlier than IJ.

To detect the key genes associated with carbohydrates during the dormancy process, gene expression levels in shoot apexes were determined in both iris species. The expression levels of genes in the starch synthesis pathway (Figure 4A), such as *SS2*, *APL2*, *ISA1* and *UGP2*, were downregulated (Figure 4B), whereas *AMY3*, *BAM1*, *BAM3*, *DPE2*, *GWD3*, *MEX1* and *SEX4*, which are in the starch catabolism pathway (Figure 5A), were upregulated in IT from dormancy induction to the dormancy stage (Figure 5B). These changes were consistent with the decline in the starch contents. During the dormancy maintenance stage, most starch synthesis and degradation genes all presented upward trends in IT, while starch content was upregulated, indicating that the rate of starch synthesis was greater than that of degradation during this stage. Nevertheless, the expression levels of starch synthesis and degradation genes fluctuated during dormancy induction in IJ.

*Invertase* (*INV*), *hexokinase* (*HK*), *sucrose synthase* (*SUS*), and *sucrose phosphate synthase* (*SPS*) play vital roles in sucrose catabolism (Figure 4A). *SPS3* was significantly upregulated at the transition from dormancy induction to the dormancy stage in IT and during dormancy induction in IJ, whereas *SUS3*, *SUS4*, *VIN1* and *VIN2* were downregulated in IT from dormancy induction to the dormancy stage. However, *SUS1*, *SUS4*, *CINV2* and *HKL1* exhibited similar patterns in IJ during dormancy induction (Figure 4B). The expression levels of sucrose catabolism genes mostly presented an increasing tendency during the dormancy maintenance stage in IT, while sucrose content was upregulated, indicating that the rate of sucrose synthesis was lower than that of degradation during this stage. Interestingly, *SUT4* which encodes a sucrose transport protein, was downregulated in IT from dormancy induction to the dormancy stage and upregulated in IJ during dormancy induction (Figure 4B).

### 2.4. Correlation of Gene Expression Analysis

Given that the present artificial dormancy induction conditions could promote the entry of complete dormancy for IT, significant correlations between genes in IT were analysed using Pearson’s correlation analysis and displayed in a network diagram (Figure 6, Appendix A). The network associates the dormancy marker genes with those involved in phytohormone biosynthesis, catabolism, signal transduction, and carbohydrate metabolism according to the expression levels. Generally, *SVP2* and *FUL2* were positively correlated with most genes, whereas *FLC1* and *FLC2* were negatively correlated with most genes. This finding was also consistent with the roles played by *SVP*, *FUL* and *FLC* during dormancy induction in several perennial plants [32,33,42]. Moreover, *SVP2* was highly correlated with genes involved in ABA signal transduction, such as *ABF1*, *BZIP12* and *PP2C5*. Other genes significantly associated with *SVP2* were *APS1*, *SS2*, *AMY3* and *BAM1*, which are key regulators of starch biosynthesis and degradation. Interestingly, *SnRK1* and *SnRK2.6* were strongly correlated with genes involved in ABA biosynthesis, catabolism, signal transduction, and carbohydrate metabolism, indicating that the crosstalk between carbohydrates and ABA might be mediated by SnRKs in irises.

## 3. Discussion

### 3.1. Early Dormancy in Deciduous Iris under Artificial Dormancy Induction Conditions

Our previous studies have shown that the winter dormancy of irises is an ecodormancy-only process [15], marked by (1) the decrease or cessation of growth; (2) senescence of above-ground foliage; and (3) reduction of metabolic activity [43]. In the transition zone between temperate and subtropical climates, evergreen irises may still maintain vegetative growth under extremely low temperatures in winter and cease growth until early February. In contrast, deciduous irises may stop vegetative growth in early autumn, and show the phenomenon of leaf wilt [15]. IT ceased growth earlier than IJ after exposure to dormancy-inducing conditions, which simulated the temperatures and light conditions in November in the Hangzhou area. We hypothesized that this finding was due to the conditions that simulated temperatures and light in November were not sufficient to induce IJ to enter dormancy completely. Consistent with this notion, IJ continued to grow in winter, and only ceased growth in early February [15]. In this regard, IT entered dormancy earlier than IJ under artificial dormancy induction conditions.

It is convenient to estimate plant dormancy status according to the expression patterns of reliable marker genes [44]. During dormancy, the shoot apex remains inactive and the expression of genes that regulate the cell cycle in plants such as *CYCD1;1* and *CDKB2;2*, are usually downregulated. Notably, *DAM*/*SVP* genes showed a significant seasonal expression pattern throughout the year, and their expression was highest during dormant bud formation or dormancy [33,45,46]. In contrast, the expression of *FLC* genes was negatively correlated with the growth cessation induced by dormancy [18,47,48]. In this study, these genes were selected as dormancy markers and their expression patterns were similar to those in the previous studies. Morphological changes of IJ that slightly increased at Tre.6 lagged behind the changes in gene transcript levels, that is, the expression of *CYCD1;1*, *CDKB2;2*, *FLC1*, and *FLC2* increased only in Tre.5 of IJ (Figure 1B,D). These results revealed that IJ was still growing at Tre.6 and the expression patterns of these marker genes were consistent with the dormancy status determination in the two iris species. Collectively, our results indicated that the artificial dormancy induction conditions were effective for irises and confirmed that the deciduous iris entered dormancy much earlier than the evergreen iris.

### 3.2. ABA and GA_3_ Coordinate Growth and Stress Response during the Dormancy of Irises

ABA plays an important role in regulating plant growth and abiotic stress [49,50]. In winter dormancy, herbaceous perennials exhibit elevated ABA levels in early autumn inducing the suppression of meristem growth [1]. Furthermore, ABA acts as a stress signal given that abiotic stresses stimulate its synthesis, especially under cold stress [51]. Similar changes in ABA levels were observed in IJ and IT, implying that ABA plays a major role in the dormancy and stress responses of irises.

Homeostasis between ABA biosynthesis and catabolism precisely regulates ABA levels in plants [6,52]. The increased ABA level noted during dormancy maintenance stage in the apical shoots of IT might be due to the remarkably increased levels of ABA synthesis-associated genes *NCED1*, *ABA2* and *AAO3* in the early stage and the decreased level of *CYP707A* (Figure 2E). The expression patterns of these ABA synthesis-associated genes and *CYP707A* were consistent with changes in ABA levels during dormancy maintenance stage in IJ (Figure 2E). In addition, these ABA synthesis-associated genes were strongly upregulated earlier in IT compared with IJ, which might contribute to the earlier entry into the dormant state in IT compared with IJ (Figure 2E). Plant growth is regulated not only by endogenous phytohormone levels but also by signal transduction pathways [53]. *PYL8*, *BZIP12*, and *ABF1* are key positive regulators in the ABA signalling pathway [12,36,54]. In our study, the above genes were upregulated during dormancy maintenance stage in IT and dormancy induction in IJ (Figure 2E). Under conditions of stress, ABA-bound PYLs interact with the conserved C-terminal catalytic domains of clade A PP2Cs, which subsequently releases the inhibition of SnRK2s by PP2Cs, thus activating stress responses [55,56]. The changes were correlated with the upregulation of *PYL8* and *SnRK*, and the downregulation of *PP2C5* throughout the entire experiment in the apical shoot tissues of IT and IJ (Figure 2E). These results indicated that the homeostasis of ABA biosynthesis and catabolism precisely regulates ABA levels and that genes in the ABA signalling pathway prominently function in dormancy induction in irises.

In contrast to ABA levels, GA levels decreased at the dormancy induction stage [11]. Similarly, decreased GA_3_ levels were observed in the apical shoot tissues of IJ during dormancy induction (Figure 2C,D). In addition, GA_3_ levels increased during ecodormancy in IT, which was consistent with results from grapes, sweet cherry, and tree peony [57,58,59]. The GA_3_ changing trends in the leaf tissues of both iris species are potentially associated with freezing tolerance [60]. With leaf withering, freezing tolerance weakened and the GA_3_ level increased in IT during the early stage of dormancy maintenance (Figure 2D). However, the freezing tolerance of evergreen IJ might be enhanced based on the decreased GA_3_ level during dormancy induction (Figure 2D). Collectively, GA_3_ coordinated plant growth and freezing tolerance during dormancy induction, whereas variations were noted depending on genetic background and organs. The DELLA protein is an important negative regulator of the GA reaction, but it is positively correlated with ABA [61]. Previous reports have indicated that DELLA proteins play an important role in dormancy and abiotic stress [62,63]. In this study, we observed that the expression of *RGA* and *GAI*, which encode the DELLA protein, was roughly correlated with ABA and GA_3_ levels (Figure 2E). These results suggested that the balancing effects of GA_3_ and ABA on dormancy and abiotic stress can be achieve by modulating DELLA.

### 3.3. Carbohydrates Actively Participate in the Dormancy Transition and Stress Responses in Irises

Carbohydrates play a very important role in plant dormancy and development [38,64] given that the conversion of starch to sugar is a key metabolic process associated with entry into dormancy [65]. Consistent with this notion, starch conversion to soluble sugars from dormancy induction to the dormancy stage was observed in the apical shoot tissues of IT (Figure 3A–D). Soluble sugars are involved in maintaining the normal function of the cell membrane, enhancing cold tolerance, and providing energy for buds at low temperatures [66,67]. The contents of soluble sugars gradually increased during dormancy maintenance stage in IT and dormancy induction in IJ in the apical shoot tissues, which may enhance their resistance to low temperatures (Figure 3B–D). Interestingly, starch content had no significant change, while soluble sugar contents showed increasing trends in the apical shoot tissues of IJ during dormancy induction (Figure 3A–D). We hypothesized that the soluble sugars in the apical shoot tissues might be derived from other tissues in IJ, and this notion is supported by the upregulation of *SUT4* during dormancy induction (Figure 4B).

The accumulation of starch in leaves after cold treatments has been observed in many herbaceous plants [68,69]. This feature might represent a special strategy for herbaceous plants to cope with cold stress. Similar changing trends of starch contents in leaves were detected in both iris species (Figure 3E). Soluble sugar contents increased during the dormancy maintenance stage in IT (Figure 3F,G), which presented opposite trends in the apical shoot tissues. This finding might be due to that leaves need to gain stronger freezing tolerance than aboveground tissue. Overall, carbohydrates act as induction signals and are involved in not only the establishment of dormancy but also the formation of protective mechanism under stress conditions.

The dormancy-growth transition process involves the extensive reconfiguration of carbohydrate metabolism as starch-derived sugars serve several purposes, for example as cryoprotectants as well as a source of energy [65,70]. Starch degradation is a multistep process involving several enzymes, and the regulatory mechanism of these enzymes during dormancy is not completely understood. In this study, *AMY3*, *BAM1* and *BAM3* were upregulated from dormancy induction to the dormancy stage in IT (Figure 5B). Similarly, *SPS3*, which is related to sucrose synthesis, was upregulated during this stage (Figure 4B). Nevertheless, the expression levels of starch synthesis and degradation genes fluctuated during dormancy induction in IJ and corresponded with no significant change in starch content (Figure 3A, Figure 4B and Figure 5B). SuSy and INV work cooperatively to cleave sucrose [71]. In the present study, the key genes regulating sucrose metabolism differed in the two iris species. The expression of *SUS3*, *CINV2* and *VIN2* showed opposite trends from dormancy induction to the dormancy stage in IT and during dormancy induction in IJ (Figure 4B). Overall, these results suggested that although carbohydrates actively participate in the dormancy transition and stress responses in both iris species, obvious differences in starch remobilization and sugar metabolism were noted between the two irises.

### 3.4. Dormancy Related MADS-box Genes and SnRKs Potentially Serve as the Bridge between Carbohydrate and Phytohormone Interaction during Iris Dormancy

*MIKC^C^ MADS-box* genes encode TFs that play crucial roles in the regulation of dormancy. To date, the role of *SVP* and *FUL* in the promotion of dormancy across different perennials has been demonstrated by numerous transgenic studies [13,33], whereas a seasonal expression analysis of *FLC* showed that these genes were downregulated during dormancy induction [18]. The expression trends of these genes during dormancy induction in the two iris species were similar to those noted in the previous studies (Figure 6), indicating the conserved function of these MIKC^C^ TFs in perennials despite distant relationships. Therefore, these *MIKC^C^ MADS-box* genes are suitable as markers to estimate the dormancy state of perennial plants.

SVP/DAM interacts with ABA to regulate winter dormancy. In poplar, the ABA signal genes and SVL form a self-enhanced regulatory loop during endodormancy. ABA enhances endodormancy and activates *SVL* expression, whereas SVL activates the expression of *NCED3* [34]. The expression of pear (*Pyrus pyrifolia* (Burm. f.) Nakai) *ABREBINDING FACTOR3* (*PpyABF3*) was positively correlated with *PpyDAM3* expression. Furthermore, PpyABF3 is directly bound to the second ABRE in the *PpyDAM3* promoter to activate its expression [12]. We observed similar results in irises. Specifically, *SVP2* was highly correlated with ABA-related genes (*ABF1*, *BZIP12* and *PP2C5*), suggesting potential conserved crosstalk between *SVP*/*DAM* and ABA during iris ecodormancy-only process. The results of gene correlation analysis also indicated possible relationships between SVP/DAM and genes responsible for starch biosynthesis and degradation. However, verified regulatory relationships among these genes are still limited and further study is needed.

Dormancy is a complex process that involves several interactions among key metabolic components including but not limited to carbohydrates and phytohormones. In this study, *SnRK1* and *SnRK2.6* were strongly correlated with genes involved in ABA biosynthesis, signal transduction, and carbohydrate metabolism. We hypothesized that the crosstalk between carbohydrates and ABA was mediated by SnRKs in irises. During bud dormancy induction of potatoes, sucrose contents decreased, and trehalose 6-phosphate (Tre6P) contents were downregulated, which activated SnRK1 activity and stimulated ABA biosynthesis [72]. Furthermore, the crosstalk between sugar and ABA can be mediated by Tre6P-SnRK1 and SNRK1-TOR (the target of rapamycin)-SnRK2 modules [73], both of which are closely related to cell division and growth [74]. These findings suggest that SnRKs may play an important role in regulating dormancy.

### 3.5. Later Growth Cessation and Higher Freezing Tolerance Potentially Result in Later Dormancy in Evergreen Iris Compared with Deciduous Iris

To obtain a more comprehensive understanding of the winter dormancy mechanism of irises, we conducted experiments on the aboveground and underground tissues of irises, namely, the apical shoot tissues and the leaf tissues. Based on a detailed comparative study on ABA, GA_3_ and carbohydrate levels, and the expression of related genes in deciduous iris IT and evergreen iris IJ under artificial dormancy induced conditions, it is believed that the earlier increase in ABA biosynthesis genes and sucrose contents in the apical shoot tissues may explain the earlier growth cessation in IT compared with IJ. Furthermore, the GA_3_ level decreased and more increasing trends in soluble sugar contents were noted in evergreen leaves under the low-temperature conditions, leading to increased freezing tolerance in IJ compared with IT. We hypothesized that later growth cessation and increased freezing tolerance resulted in later dormancy of IJ compared with IT (Figure 7).

## 4. Materials and Methods

### 4.1. Plant Materials and Treatments

Two one-year-old iris species, the deciduous *I. tectorum* and the evergreen *I. japonica*, bought from a local nursery, were used in this experiment. To ensure the good consistency of the experimental materials, two iris species growing uniformly by vegetative propagation were planted in a 3:1 (*v*/*v*) mixture of peat and vermiculite in October. Afterwards, these plants were grown in the Perennial Flower Resources Garden of Zhejiang University under natural conditions and managed conventionally in Hangzhou (30°15′ N, 120°10′ E), China. After one month, 30 pots of uniformly sized plants for each iris species were moved to chambers (12 h light/12 h dark, 60 μmol photons m^−2^ s^−1^, 55% relative humidity) at 4 °C for one week to allow for cold acclimation. Then these plants were transferred into dormancy-inducing conditions in a climate chamber (55% relative humidity), which simulates natural air temperatures and photoperiod in November in Hangzhou. The conditions of the climate chamber are shown in Table 1.

According to the growth status of the two iris species, six time points with significant morphological differences were selected for measuring and sampling, namely, the day of transfer into the dormancy-inducing chamber, the 3rd day after transferring (DAT), the 12th DAT, the 15th DAT, the 18th DAT, and the 30th DAT. These time points were recorded as Tre.1, Tre.2, Tre.3, Tre.4, Tre.5 and Tre.6, respectively, for convenience (Figure 1A). On each sampling day, apical shoots (including the shoot apex and three youngest leaves) and functional leaves (including the third and fourth actively growing leaves from the central part to both sides in the plant) were collected at 14.00~15.00 h, flash frozen in liquid nitrogen and stored at −80 °C until further use.

### 4.2. Determination of the Dormancy of Plants

To estimate the dormancy status of the plants sampled, we randomly measured changes in the shoot height and the number of green leaves (greater than 50% of the leaves are green) of nine plants at six sampling stages for the two iris species (Figure 1A). The two species were clipped at a height of 10.00 cm above the soil surface to measure plant growth clearly when moved to the climate chamber. Plants without significant changes in shoot height were considered to be dormant. The iris homologues to *Arabidopsis thaliana CYCD1;1*, *CDKB2;2*, *SVP*, *FLC* and *FUL* were also used as dormancy markers according to the literature [44].

### 4.3. Determination of Carbohydrate Contents

Based on the changes in growth status, samples from Tre.1, Tre.4, Tre.5 and Tre.6 were selected for the determination of carbohydrate contents (Figure 1A). The method employed for the extraction and quantification of fructose, glucose, and sucrose was described by Fan et al., 2019 [75]. Specifically, 20 mg of the collected sample was ground in powder by liquid nitrogen, extracted with 5 mL 80% (*v*/*v*) ethanol for 60 min at 80 °C centrifuged at 13,000× *g* for 5 min. Then, the supernatants were collected. Alcohol extraction was repeated two additional times under the same conditions. Extracts for each sample were pooled, clarified by adding approximately 1 g of activated charcoal for 2 h and evaporated to dryness in 45 °C oven. The residue was resuspended in 1 mL of ultrapure water and filtered through sterile filters (cut off size 0.22 μm). The sugar content measurements were performed by high-performance liquid chromatography (Agilent 2100 system; Palo Alto, CA, USA) as described by Ma et al., 2017 [76]. The starch content was determined after soluble sugar extraction using the anthrone method [77] according to Wang et al., 2014 [78].

### 4.4. Measurement of Endogenous Phytohormone Levels

Samples from Tre.1, Tre.4, Tre.5 and Tre.6 were selected for the determination of endogenous phytohormone levels (Figure 1A), which were determined by enzyme-linked immunosorbent assay (ELISA) as described previously by Chen et al., 2006 [79]. Specifically, frozen samples were ground to a fine powder and extracted in 4 mL of 80% (*v*/*v*) methanol containing 1 mmol L^−1^ butylated hydroxytoluene to prevent oxidation. The resulting extract was incubated for 24 h at 4 °C and then centrifuged at 6300× *g* (4 °C, 10 min). The supernatant was filtered through Chromosep C18 columns (C18 Sep-Park Cartridge; Waters) that were prewashed with 10 mL of 100% (*w*/*v*) and 5 mL of 80% (*v*/*v*) methanol. Two milliliters of the eluted phytohormone fractions were pooled and dried in a stream of N2. Then the residue was dissolved in 0.5 mL sodium phosphate buffer containing 0.1% (*v*/*v*) Tween 20 and 0.1% (*w*/*v*) gelatine (pH 7.5). Endogenous phytohormone levels were quantified by ELISA with the standards for ABA and GA_3_ (Yuanye Biochemical Company, Shanghai, China), following the manufacturer’s protocol.

### 4.5. Quantitative Real-Time PCR

To explore the expression patterns of genes related to pathways that play important roles in the regulation of plant dormancy, total RNA was extracted from the apical shoot of the two iris species sampled on Tre.1, Tre.4, Tre.5 and Tre.6 by a total RNA extraction kit (Tiangen, Beijing, China). RNA samples were photometrically quantified and verified on an agarose gel. A PrimeScript™ RT Reagent Kit with gDNA Eraser (TaKaRa, Kyoto, Japan) was used for cDNA synthesis following the manufacturer’s protocol. Candidate genes were selected from the literature and the differentially expressed transcripts (DETs) were selected based on our previous transcriptome analyses of IT and IJ during natural winter dormancy [15]. Primers for quantitative real-time PCR (qRT–PCR) were designed according to the iris transcriptome [15] and verified in the two iris species by clone sequencing, with an expected product fragment size of 100–200 bp and a similar melting temperature (Tm) value (approximately 60 °C) for each primer. Primer sequences are listed in Appendix A. The cDNA was run on a CFX Connect™ Real-Time PCR Detection System (Bio-Rad, Hercules, CA, USA) using TB Green^®^ Premix Ex Taq (TaKaRa, Kyoto, Japan). *Actin* was used as the internal control because it was stably expressed according to iris RNA sequencing [15]. The following PCR protocol was employed: 2 min at 95 °C; 39 cycles of 5 s at 95 °C and 30 s at 55 °C; and a melting curve program of 5 s at 95 °C, 5 s at 65 °C and 5 s at 95 °C. Three biological replicates were used for each analysis. The relative expression levels of these genes were determined using the 2^−ΔΔCt^ method as described by Livak and Schmittgen, 2001 [80].

### 4.6. Statistical Analyses

All experiments in this study were conducted following a completely randomized design. One-way analysis of variance (ANOVA) was used to compare differences among different indices or treatments via SPSS 26.0 (IBM Corp., Armonk, NY, USA), with a probability value of *p* < 0.05 considered significant. Correlation analyses between gene expression data in IT were performed using Pearson’s two-tailed tests. GraphPad Prism 9.0 (GraphPad Software, Inc., La Jolla, CA, USA), Tbtools v1.046 (Guangzhou, China) [81], and Cytoscape v3.9.1 (Paul Shannon, Seattle, WA, USA) [82] were applied for visualization of the experimental data.

## 5. Conclusions

Dormancy is a protective survival strategy for plants to resist harsh natural environments [1]. However, the underlying molecular mechanisms of winter dormancy are incompletely understood, especially in perennials with only ecodormancy. Here, we found that the deciduous iris entered dormancy earlier than the evergreen iris under artificial induction dormancy conditions. Phytohormones and carbohydrates played roles in coordinating growth and stress response during dormancy induction in both iris species. These findings also demonstrated to some extent that the winter dormancy of irises is ecodormancy. Furthermore, dormancy related *MADS-box* genes and *SnRKs* might serve as the bridge between carbohydrate and phytohormone interaction during iris dormancy. These findings provide a hypothetical model of the reason for later dormancy in evergreen iris compared with deciduous iris under artificially induced dormancy conditions and reveal some candidate genes. These findings could provide new insights into the research of dormancy in perennial plants with an ecodormancy-only process and contribute to the effective management of iris production, postharvest storage, and shipping. Further studies should aim to determine the interactions between different dormancy response pathways and to comprehensively and reliably identify the functional and transcriptional networks of dormancy candidate genes. In addition, it is challenging to thoroughly investigate new genes or mechanisms related to the dormancy process in irises.

## Figures and Tables

**Figure 1 ijms-23-11123-f001:**
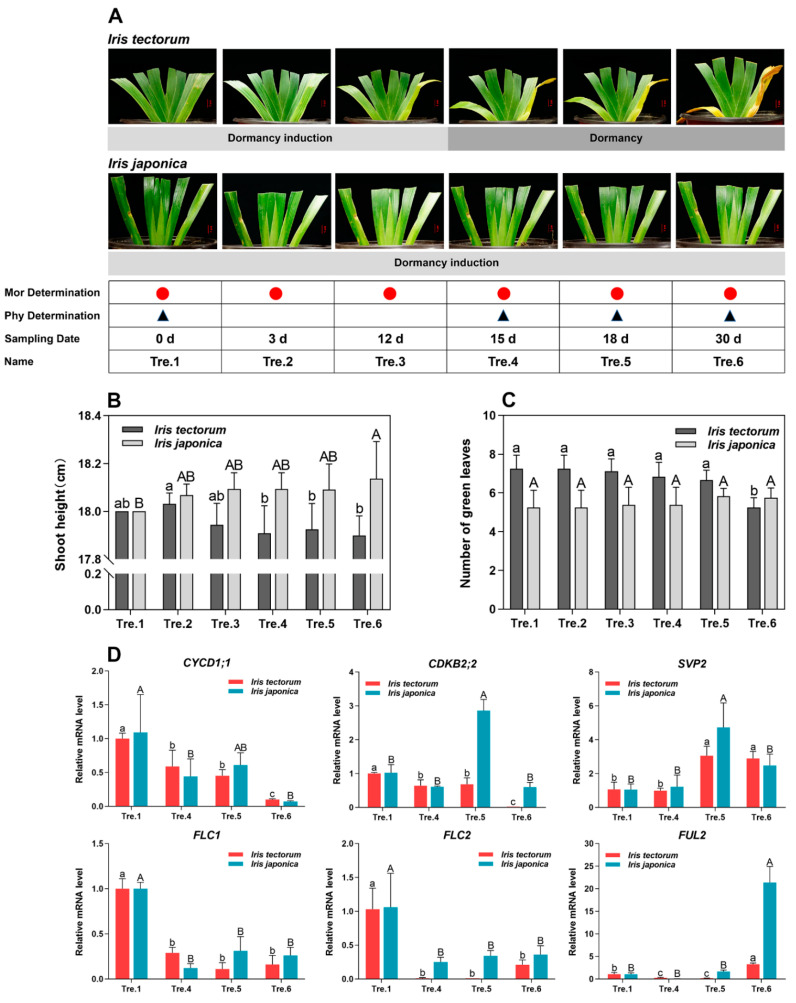
Sample harvesting, morphological observations, and the expression changes of marker genes in this study. (**A**) Plant status of *Iris tectorum* (IT) and *I. japonica* (IJ) at the stages for morphological observations, sampling dates for morphological (Mor) observations and physiological (Phy) measurements, and the corresponding dormancy status as defined in this study. Red circles represent plants sampled for morphological observations, and black triangles represent plants sampled for physiological measurements. (**B**,**C**) Morphological observations of the two iris species. (**B**) Shoot height (cm). (**C**) The number of green leaves. (**D**) Relative mRNA levels of marker genes in both iris species. Gene full names are shown in Appendix A. The data in (**B**,**C**) are means from nine biological replicates and the data in (**D**) are means from three biological replicates, with error bars representing standard deviation. The lowercase and uppercase letters denote significant differences for relevant parameters within IT and IJ (Duncan’s multiple comparison test *p* < 0.05), respectively.

**Figure 2 ijms-23-11123-f002:**
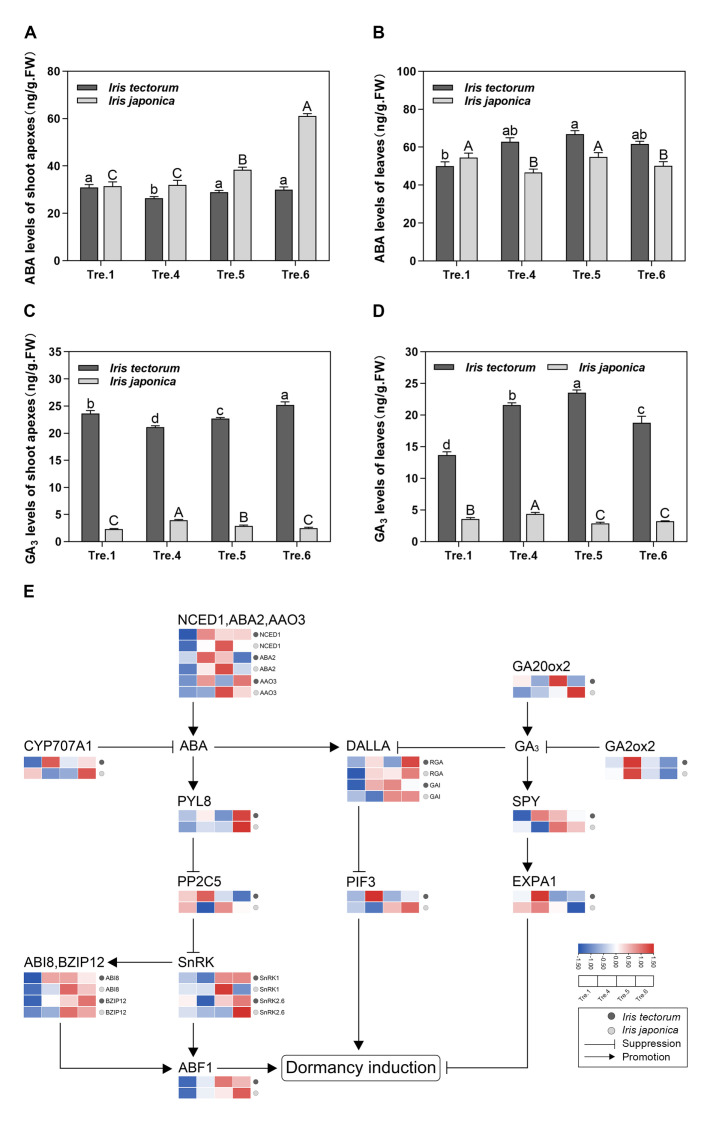
Changes in levels of ABA and GA_3_ and transcript expression of genes associated with ABA and GA_3_ pathways in *Iris tectorum* (IT) and *I. japonica* (IJ), under artificial dormancy induction conditions. (**A**) ABA level changes of shoot apexes in the two iris species. (**B**) ABA level changes of leaves in the two iris species. (**C**) GA_3_ level changes of shoot apexes in the two iris species. (**D**) GA_3_ level changes of leaves in the two iris species. The data in (**A**–**D**) are means from three biological replicates, with error bars representing standard deviation. Lowercase and uppercase letters represent significant differences for relevant parameters within IT and IJ (Duncan’s multiple comparison test *p* < 0.05), respectively. (**E**) Transcript expression of genes associated with ABA and GA_3_ pathways in IT and IJ under artificial dormancy induction conditions. Red denotes that the gene is upregulated, whereas blue represents that the gene is downregulated. Gene full names are shown in Appendix A.

**Figure 3 ijms-23-11123-f003:**
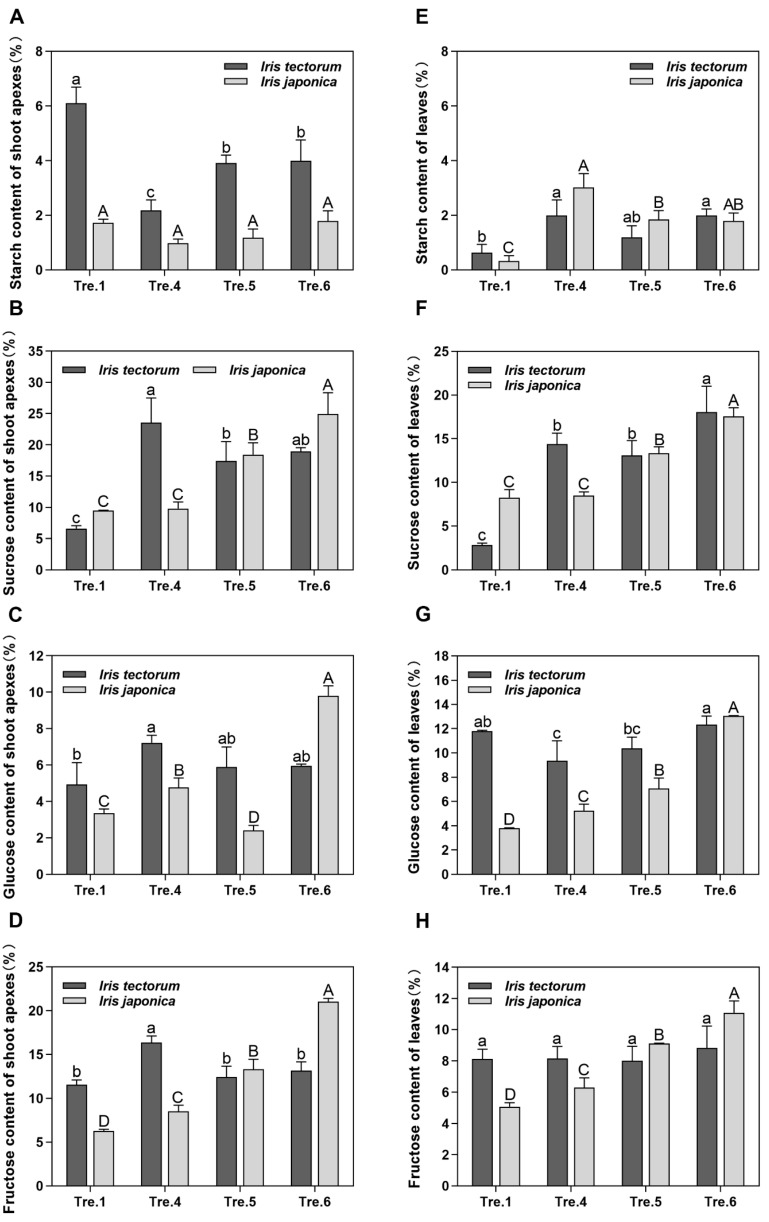
Changes in carbohydrate content in shoot apexes and leaves of *Iris tectorum* (IT) or *I. japonica* (IJ) under artificial dormancy induction conditions. (**A**) Starch content changes of shoot apexes in the two iris species. (**B**) Sucrose content changes of shoot apexes in the two iris species. (**C**) Glucose content changes of shoot apexes in the two iris species. (**D**) Fructose content changes of shoot apexes in the two iris species. (**E**) Starch content changes of leaves in the two iris species. (**F**) Sucrose content changes of leaves in the two iris species. (**G**) Glucose content changes of leaves in the two iris species. (**H**) Fructose content changes of leaves in the two iris species. The data in (**A**–**H**) are means from three biological replicates, with error bars representing standard deviation. Lowercase and uppercase letters represent significant differences for relevant parameters within IT and IJ (Duncan’s multiple comparison test *p* < 0.05), respectively.

**Figure 4 ijms-23-11123-f004:**
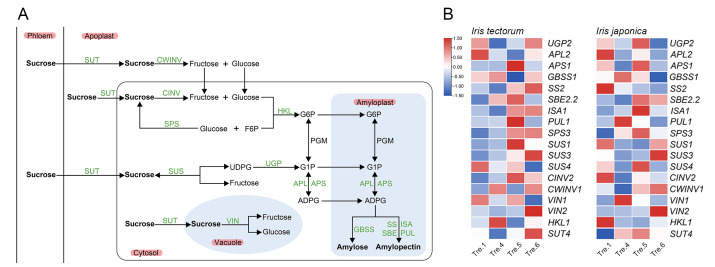
Transcript expression of genes associated with sucrose metabolism and starch synthesis pathways in *Iris tectorum* (IT) and *I. japonica* (IJ), under artificial dormancy induction conditions. (**A**) A schematic illustration of sucrose metabolism and starch synthesis pathways. (**B**) Transcript expression of genes associated with sucrose metabolism and starch synthesis pathways in IT and IJ under artificial dormancy induction conditions. Red denotes that the gene is upregulated, whereas blue represents that the gene is downregulated. Gene full names are shown in Appendix A.

**Figure 5 ijms-23-11123-f005:**
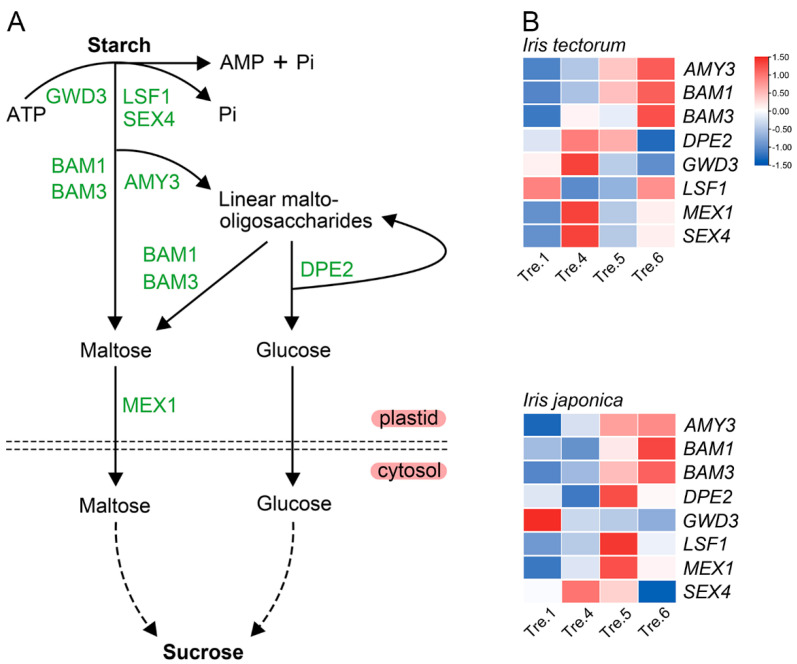
Transcript expression of genes associated with starch degradation pathways in *Iris tectorum* (IT) and *I. japonica* (IJ), under artificial dormancy induction conditions. (**A**) A schematic illustration of starch degradation pathways. (**B**) Transcript expression of genes associated with starch degradation pathways in IT and IJ under artificial dormancy induction conditions. Red denotes that the gene is upregulated, whereas blue represents that the gene is downregulated. Gene full names are shown in Appendix A.

**Figure 6 ijms-23-11123-f006:**
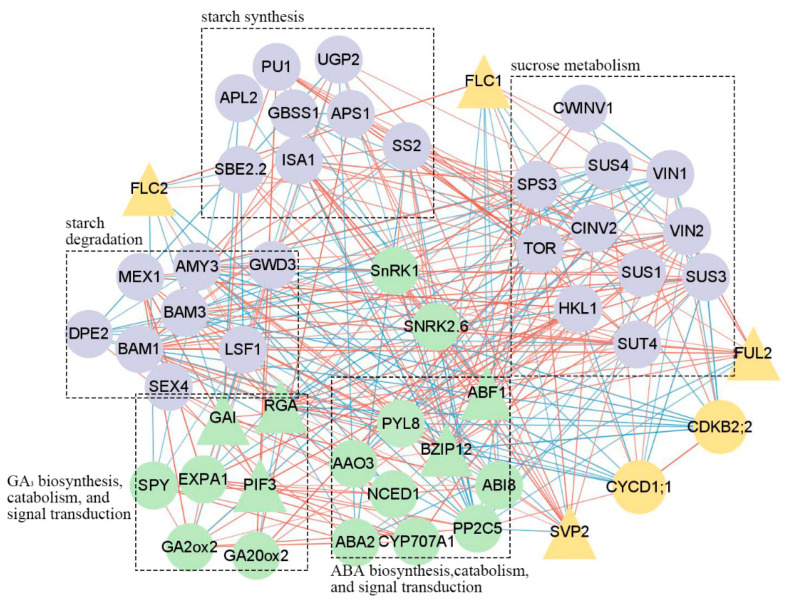
The gene correlation analysis in *Iris tectorum* during artificial dormancy induction. The correlation was analysed using a Pearson’s two-tailed test. Genes with significant correlations (*p* < 0.05) were linked together. The thicker the line is, the correlation coefficient is larger. The red line indicates a positive correlation, and the blue line indicates a negative correlation. Genes with yellow background represent marker genes, genes with green background represent genes involved in the phytohormone biosynthesis, catabolism, and signal transduction, and genes with purple background represent genes involved in the carbohydrate metabolism. Circles represent functional genes, and triangles represent transcription factors. Gene full names are shown in Appendix A.

**Figure 7 ijms-23-11123-f007:**
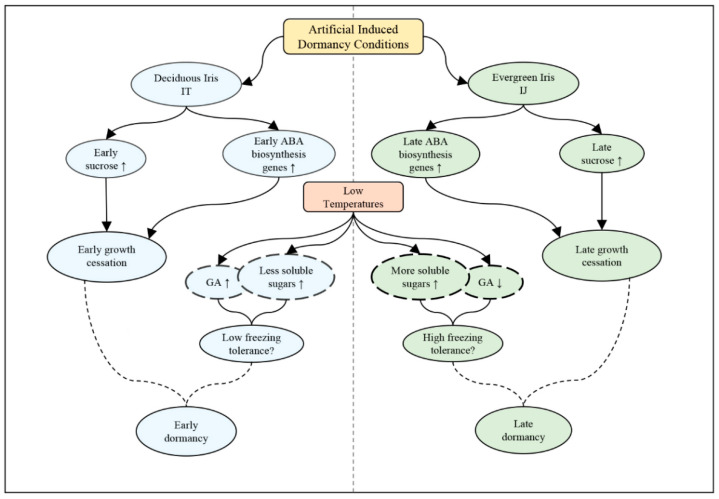
Proposed model of the reason for later dormancy in evergreen iris compared with deciduous iris under artificial induced dormancy conditions. Blue represents IT, and the green represents IJ. Solid wireframes represent physiological and molecular responses in apical shoot tissues, dotted wireframes represent physiological and molecular responses in leaf tissues.

**Table 1 ijms-23-11123-t001:** The artificial dormancy induction conditions.

Temperature (°C)	Processing Time (h)	Time Distribution	Light Intensity (μmol Photons m^−2^ s^−1^)
0	1.0	0:00–5:00	0
3	2.0	5:00–7:00	0
3	2.0	7:00–9:00	60
7	5.0	9:00–14:00	60
4	3.5	14:00–17:30	60
4	0.5	17:30–18:00	0
0	1.0	18:00–24:00	0

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
