# Peer review of "Later Growth Cessation and Increased Freezing Tolerance Potentially Result in Later Dormancy in Evergreen Iris Compared with Deciduous Iris"

_ijms, 2022, doi:10.3390/ijms231911123_

Round 1
Reviewer 1 Report
It is a well-planned and well-presented piece of research work. A comprehensive story of ecodormancy in deciduous and evergreen iris is studied, covering its physiological, biochemical, and molecular aspects.
A few typos are spotted/highlighted in the attached PDF.
The below studies can be checked for plagiarism/similarities:
Li, D., Shao, L., Zhang, J., Wang, X., Zhang, D., Horvath, D.P., Zhang, L., Zhang, J. and Xia, Y., 2022. MADS-box transcription factors determine the duration of temporary winter dormancy in closely related evergreen and deciduous Iris spp. Journal of Experimental Botany, 73(5), pp.1429-1449.
Dole, J. M. (2003). Research approaches for determining cold requirements for forcing and flowering of geophytes. HortScience, 38(3), 341-346.

Reviewer 2 Report
The studies show in a very elegant way the stages that culminate in dormancy and its awakening in two apparently important plant species. The study was very well conducted, with excellent results and graphics that enhance the study. The authors are to be congratulated for the excellent images they present in the analyzed material. I believe that the construction of the correlation matrix must be a lot of work, I saw it and I was impressed; once again congratulations to the authors. Just as there is no marked transition from the vegetative phase to the dormant phase, there are not only flowers in this garden either. Minor flaws have occurred and should be reviewed. Below I mention the main
Introduction
1) There is a very clear rule accepted worldwide for the description of species and this cannot be ignored as was done in line 86. It is a consensus among botanists that in the case of many species or with very complicated names, a list should be made containing the common name of the plant, its scientific name and that it be assigned a number which will be followed throughout the text. As in the present study there are only two species of the same genus, the official nomenclature must be adopted. This recommendation applies to the entire manuscript.
2) Lines 92-94: “In autumn, increased.... onset of dormancy” these two sentences must be joined into one to become a cohesive text
3) 3) Lines 94-95: As a senior student of seed germination I do not agree with the information that carbohydrates are the main reserve source. Seeds are generally classified as starchy: as glucose units join together to form starch (eg. corn), oilseeds: where the main form of storage of reserves occurs from lipids (peanuts) and proteins: where the main source of reserves are proteins (soybean). Another mistake in this sentence is the citation of an article about Arabidopsis, a plant with very small seeds and poor in reserves. I suggest rethinking the phrase and the reference
4) Lines 94-97): The phrase “Of these, soluble sugars are involved in maintaining the normal function of cell membranes, enhancing cold tolerance and acting as signal molecules regulating plant growth” suggests that soluble sugars, that is, reducing sugars, are used as a reserve. If so, the authors should study a little more plant physiology, because reducing sugars would hardly be a reserve source, precisely because of their ease of entering the respiratory tract of the “dormant” seed and then depleting the reserves before germination. But if that was not the focus of the sentence, then it seems to me a little out of focus. Check that in lines 404-407 the correct order is maintained, while in the previous text it was quite confusing. Double check pl.
5) Line 103: I didn't understand the difference between SVP and SVL, it was not clear
Materials and Methods
6) Why determination of carbohydrate content were made in only Tre.1, Tre.4, Tre.5, and Tre.6? The same question is valid to line 551, 567
7) Line 560. Basically, there are two types of phosphate buffer: sodium phosphate buffer and potassium phosphate buffer, as it is only referred to as phosphate-buffered saline it is not clear which buffer is used because potassium is also a salt as well as sodium
Results
8) Line 137, the citation of table 1 is made on page 3 and the table only appears on page 15, this cannot happen, the figure or table where the data are located must be close to the place where it is called, because that way makes viewing easier.
9) Figure 1. Why Iris tectorum always presents a very high standard deviation represented as the square of the standard error presented?
10) “According to morphological observations and the expression of marker genes, Tre.1 to Tre.3 were classified as the dormancy induction stage, whereas Tre.4 to Tre.6 were considered the dormancy maintenance stage for IT.” I don't understand the logic of this thought, please be more clear
11) 11) As IJMS journal uses the criterion of presenting the materials and methods at the end of the paper, the sequence for calling tables and figures must be followed from page 1 to page k. This was noticed with Table 1 and with Figure 1A. In the case of Figure 1A, it appears in the text long before it is cited, unlike Table 1, which is only presented 12 pages later. I believe this should be corrected for manuscript fluency
12) Figure 1: There is a big difference between “error bars” and “standard deviation”. These are two important statistical considerations that the authors must not confuse. So, what is being presented in Figure 1 is the “error bars” or the “standard deviation”? The same occurs in fig 2-4
13) Line 166. It is not common in the results section to present a reference citation. This author's statement should be translocated to the discussion section or simply deleted. The same happens on line 198
14) In Figure 4A, there is a consensus among plant physiologists that cytoplasm is everything that is inside the plasma membrane where the organelles are embedded, the same would be true for the cytosol. However, in plant cells 50% to 90% of its volume is occupied by the vacuole, so for plant cells, it is called cytoplasm when considering the vacuole or cytosol when it is discounted. So, I believe that the best word, in this case, would be cytosol and not cytoplasm as in animal cells.
Discussion
15) At various times I felt that the discussion was more like a restatement or continuation of the presentation of the results than a true discussion. Just as it is not common to cite references in the results section, it is not common to call figures and tables in the discussion section, since the tables and figures should already have been well presented in the results section. To me, that sounds a little strange.
16) The discussion section is very long and tiring to read. Several times I felt like I was reading a review rather than a discussion of a scientific study. I believe that the discussion can be improved by removing the restatement of the data and a better discussion of the data presented in the study and not as if it were a literature review or an introduction.
Author Response
Response to Reviewer #2
The manuscript number is: ijms-1908891
Overall comments
The studies show in a very elegant way the stages that culminate in dormancy and its awakening in two apparently important plant species. The study was very well conducted, with excellent results and graphics that enhance the study. The authors are to be congratulated for the excellent images they present in the analyzed material. I believe that the construction of the correlation matrix must be a lot of work, I saw it and I was impressed; once again congratulations to the authors. Just as there is no marked transition from the vegetative phase to the dormant phase, there are not only flowers in this garden either. Minor flaws have occurred and should be reviewed. Below I mention the main
Response:
We are truly grateful for Reviewer #2’s affirmation on this study topic and it is very important for further studies. Winter dormancy is a protective survival strategy for plants to resist harsh natural environments. However, little is known about the mechanism underlying this process in irises, which only exhibit ecodormancy during the overwintering process in the transition zone between temperate and subtropical climates. This makes us feel unsure about the prospect of this study even though we have a number of interesting findings. Your support gave us abundant confidence to continue this study topic. We’ll try our best to modify and improve this manuscript.
Comment 1
Introduction
1) There is a very clear rule accepted worldwide for the description of species and this cannot be ignored as was done in line 86. It is a consensus among botanists that in the case of many species or with very complicated names, a list should be made containing the common name of the plant, its scientific name and that it be assigned a number which will be followed throughout the text. As in the present study there are only two species of the same genus, the official nomenclature must be adopted. This recommendation applies to the entire manuscript.
Response:
We do agree with Reviewer #2 that the official nomenclature should be used for the description of species. When Iris japonica Thunb. and Iris tectorum Maxim. appeared for the first time in this paper (line 72-73 of revised manuscript): ‘Iris japonica Thunb. (IJ) and Iris tectorum Maxim. (IT) are closely related species’, we adopted the official nomenclature. For the convenience of description, we adopted the abbreviations of scientific names of Iris japonica Thunb. and Iris tectorum Maxim., namely IJ and IT, in the following text. This is a common usage in most scientific papers.
2) Lines 92-94: “In autumn, increased.... onset of dormancy” these two sentences must be joined into one to become a cohesive text
Response:
Thank you for pointing this out. We have rephrased these two sentences in the revised manuscript according to the suggestion of Reviewer #2: In autumn, increased ABA levels and decreased gibberellin (GA) levels result in growth cessation, apical buds set, and induction of bud dormancy.
3) Lines 94-95: As a senior student of seed germination I do not agree with the information that carbohydrates are the main reserve source. Seeds are generally classified as starchy: as glucose units join together to form starch (eg. corn), oilseeds: where the main form of storage of reserves occurs from lipids (peanuts) and proteins: where the main source of reserves are proteins (soybean). Another mistake in this sentence is the citation of an article about Arabidopsis, a plant with very small seeds and poor in reserves. I suggest rethinking the phrase and the reference
4) Lines 94-97): The phrase “Of these, soluble sugars are involved in maintaining the normal function of cell membranes, enhancing cold tolerance and acting as signal molecules regulating plant growth” suggests that soluble sugars, that is, reducing sugars, are used as a reserve. If so, the authors should study a little more plant physiology, because reducing sugars would hardly be a reserve source, precisely because of their ease of entering the respiratory tract of the “dormant” seed and then depleting the reserves before germination. But if that was not the focus of the sentence, then it seems to me a little out of focus. Check that in lines 404-407 the correct order is maintained, while in the previous text it was quite confusing. Double check pl.
Response:
Sorry for the confusion caused by these sentences. We have modified them in lines 95-100 and lines 412-418 of the revised manuscript. New citations have been added in the modified sentence.
5) Line 103: I didn't understand the difference between SVP and SVL, it was not clear
Response:
Sorry for the confusion caused by this sentence. SVL (short vegetative phase-like) is a MADS-box gene homologous to SVP (short vegetative phase) in Arabidopsis [1]. We have modified it in the revised manuscript, specifically, in line 104-106 of revised manuscript: ‘In poplar (Populus tremula L.), ABA promotes the expression of short vegetative phase-like (SVL), which subsequently positively regulates ABA synthesis’.
Comment 2
Materials and Methods
6) Why determination of carbohydrate content were made in only Tre.1, Tre.4, Tre.5, and Tre.6? The same question is valid to line 551, 567
Response:
To improve the experimental efficiency and save costs, only samples of key developmental stages were selected for detailed physiological and molecular analyses. Notably, obvious differences in the growth status of the two iris species were noted since Tre.4. The shoot height of IT tended to be stable, indicating a state of growth cessation, whereas IJ was still in a slow-growth trend at a growing state. Therefore, Tre.1, Tre.4, Tre.5, and Tre.6 were selected for the determination of carbohydrate contents and endogenous phytohormone levels. We also explained this in the paper: ‘To further dissect the differences between evergreen and deciduous irises during dormancy, sampling stages when morphological differences occurred (Tre.1, Tre.4, Tre.5, and Tre.6), were selected for further experiments’.
7) Line 560. Basically, there are two types of phosphate buffer: sodium phosphate buffer and potassium phosphate buffer, as it is only referred to as phosphate-buffered saline it is not clear which buffer is used because potassium is also a salt as well as sodium
Response:
Thank you for pointing this out. In the experiment, we used sodium phosphate buffer. We have modified this in the revised manuscript.
Comment 3
Results
8) Line 137, the citation of table 1 is made on page 3 and the table only appears on page 15, this cannot happen, the figure or table where the data are located must be close to the place where it is called, because that way makes viewing easier.
11) As IJMS journal uses the criterion of presenting the materials and methods at the end of the paper, the sequence for calling tables and figures must be followed from page 1 to page k. This was noticed with Table 1 and with Figure 1A. In the case of Figure 1A, it appears in the text long before it is cited, unlike Table 1, which is only presented 12 pages later. I believe this should be corrected for manuscript fluency
Response:
The above error has been revised. Similar errors have been checked and modified as well in the revised manuscript.
9) Figure 1. Why Iris tectorum always presents a very high standard deviation represented as the square of the standard error presented?
Response:
The reasons are as follows:
(1) In the morphological observation, we used biological duplication. Even if the size of the plants used in our experiment was as consistent as possible, individual differences would still lead to some differences in shoot height after a period of growth.
(2) The standard deviations in Figure 1 appear high because the ordinate is concentrated between 17.8 and 18.2 cm to highlight changes in shoot height during dormancy induction.
10) “According to morphological observations and the expression of marker genes, Tre.1 to Tre.3 were classified as the dormancy induction stage, whereas Tre.4 to Tre.6 were considered the dormancy maintenance stage for IT.” I don't understand the logic of this thought, please be more clear
Response:
For morphological observations, the shoot height decreased gradually and then stabilized over time in IT, whereas that of IJ increased slowly (Fig. 1B). Obvious differences in the growth status of the two iris species were noted since Tre.4. The shoot height of IT tended to be stable, indicating a state of growth cessation, whereas IJ was still in a slow-growth trend at a growing state.
In both species, the expression of CYCD1;1, FLC1, and FLC2 decreased during the dormancy-inducing conditions, whereas SVP2 and FUL2 expression increased (Fig. 1D). The transcript levels for CYCD1;1, CDKB2;2, FLC1, and FLC2 increased from Tre.4 to Tre.5 in IJ, suggesting that IJ still had the tendency to grow at this stage. Thus, Tre.1 to Tre.3 were classified as the dormancy induction stage, and Tre. 4 to Tre.6 were the dormancy maintenance stage for IT, while IJ was at the dormancy induction stage from Tre.1 to Tre.6.
12) Figure 1: There is a big difference between “error bars” and “standard deviation”. These are two important statistical considerations that the authors must not confuse. So, what is being presented in Figure 1 is the “error bars” or the “standard deviation”? The same occurs in fig 2-4
Response:
Error bars can be used to indicate standard deviation or standard error. In this paper, error bars represent standard deviation in Fig.1-4. We also illustrate this in the caption in Fig. 1-4: ‘with error bars representing standard deviation’.
13) Line 166. It is not common in the results section to present a reference citation. This author's statement should be translocated to the discussion section or simply deleted. The same happens on line 198
Response:
We have removed the reference citations and related statements in the results section as Reviewer #2’s suggestion.
14) In Figure 4A, there is a consensus among plant physiologists that cytoplasm is everything that is inside the plasma membrane where the organelles are embedded, the same would be true for the cytosol. However, in plant cells 50% to 90% of its volume is occupied by the vacuole, so for plant cells, it is called cytoplasm when considering the vacuole or cytosol when it is discounted. So, I believe that the best word, in this case, would be cytosol and not cytoplasm as in animal cells.
Response:
Thank you for pointing this out. We have revised Figure 4A and modified ‘cytoplasm’ into ‘cytosol’ in the revised manuscript.
Comment 4
Discussion
15) At various times I felt that the discussion was more like a restatement or continuation of the presentation of the results than a true discussion. Just as it is not common to cite references in the results section, it is not common to call figures and tables in the discussion section, since the tables and figures should already have been well presented in the results section. To me, that sounds a little strange.
16) The discussion section is very long and tiring to read. Several times I felt like I was reading a review rather than a discussion of a scientific study. I believe that the discussion can be improved by removing the restatement of the data and a better discussion of the data presented in the study and not as if it were a literature review or an introduction.
Response:
According to the suggestion of Reviewer #2, we have deleted some unnecessary information in the discussion section and reworded the discussion of some research data to make the revised manuscript more concentrated and concise.
References
- Azeez, A.; Zhao, Y. C.; Singh, R. K.; Yordanov, Y. S.; Dash, M.; Miskolczi, P.; Stojkovic, K.; Strauss, S. H.; Bhalerao, R. P.; Busov, V. B. EARLY BUD-BREAK 1 and EARLY BUD-BREAK 3 control resumption of poplar growth after winter dormancy. Nature Communications 2021, 12 (1).